# Brain Tumor Segmentation using Topological Loss in Convolutional Networks

**Charan Reddy**                                                    RAGIREDDYCHARAN223@GMAIL.COM
**Karthik Gopinath**                                               KARTHIK.GOPINATH.1@ETSMTL.NET
**Herve Lombaert**                                                  HERVE.LOMBAERT@ETSMTL.CA
*ETS Montreal, Canada*

## Abstract

Fully Convolutional Networks (FCNs) are widely used in medical image analysis for segmentation tasks. However, most FCNs fail to directly incorporate image geometry such as topology and boundary smoothness during segmentation. The sub-regions of the brain tumor in MRI images follow a particular topological order. However, the automatic segmentation of these brain tumors with conventional FCNs may violate the topological structure of brain tumors. FCNs could be constrained with a topological loss to enforce structured predictions. This paper presents the effect of such topological loss on brain tumor segmentation using the BraTS dataset.

**Keywords:** Brain Tumor Segmentation, Topology, Image Segmentation, Deep Learning

## 1. Introduction

Among brain tumors, gliomas directly impact expectancy and quality of life. Depending on their aggressiveness and histological heterogeneity, the grades I-II are termed as low-grade gliomas and the grades III-IV as high-grade gliomas (Kleihues et al., 1993; Louis et al., 2007). Modern MRI techniques reveal features for detecting such tumors in T1, T2, T1C or FLAIR images (Bakas et al., 2017). Present methods of treatment for brain tumors include surgery, radiotherapy or chemotherapy. Radiotherapy uses image segmentation by focusing radiation on segmented brain tumors. However, current methods are limited by the availability of manual segmentation in MRI images. Automating such tedious manual task is therefore highly sought (Litjens et al., 2017; Menze et al., 2015).

Current methods for brain tumor segmentation include generative and discriminative models. Generative models use prior information about the healthy tissue or expected shape of the tissue to segment an image. The challenge of such probabilistic models resides in exploiting expected shapes in images. Discriminative models learn image features to classify tissues. These models, however, require calibrated image intensities across all input images. Neural networks are discriminative models where features are extracted by the network itself. However, shape priors are often ignored when modeling specific tissues. The approaches based on Cascaded FCNs (Christ et al., 2016; Wang et al., 2017) also ignore topological constraints. Hence, we explore how the use of a topological loss (BenTaieb and Hamarneh, 2016) can penalize invalid topology when segmenting brain tumor structures.

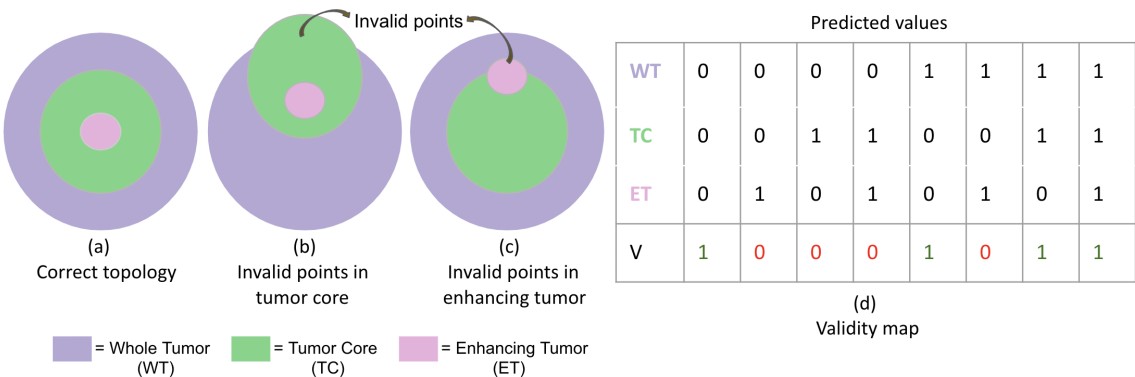

Figure 1: Case (a) shows correct topology, while (b) and (c) have regions deviating from topology. Figure (d) presents the validity map $V$ for the predicted vector.

## 2. Method

The aim of this paper is to segment various subregions of a brain tumor, namely whole tumor (WT), tumor core (TC) and enhancing tumor (ET). Cascaded FCNs perform multiple binary class segmentation instead of multi-class segmentation (Christ et al., 2016). Such training does not ensure the removal of tumor core outliers beyond the WT or ET regions.

Therefore, we explore the use of a topological loss $L_{Topo}$ (BenTaieb and Hamarneh, 2016) in addition to dice loss $L_{Dice}$ to enforce the desired voxel topology.

$$L = L_{Dice} + L_{Topo} + L_{Smooth} \tag{1}$$

$$L_{Topo}(x;\Theta) = \sum_{p \in \Omega} \sum_{r=1}^{R} -y_p^r \times logP(y_p^r|x_p;\Theta) \times V,$$

$$\forall y_p \in \{0,1\}^R \tag{2}$$

Figure 1 gives a binary validity map $V$ for each pixel $p$. $x_p$ and $y_p$ are the voxel and ground truth segmentation values. $P$ are the class probability outputs from softmax after FCN. $R$ denotes subregions of the tumor. $V$ will be 1 if the predicted value of pixel follows the topology otherwise, the value is 0. We have also added a smoothing loss function $L_{Smooth}$ (BenTaieb and Hamarneh, 2016) to enforce regularization in the segmentation boundaries.

## 3. Experiments and Results

Cascaded FCNs trained with a Dice loss serve as our baseline method (Wang et al., 2017). In the first experiment, we add a topology loss in addition to the Dice loss. In the second experiment, we enforce regularization by adding a smoothing term to the previous Dice and topology loss. Our training and testing sets are respectively split using 250 and 35 images from the BRATS dataset (Menze et al., 2015).

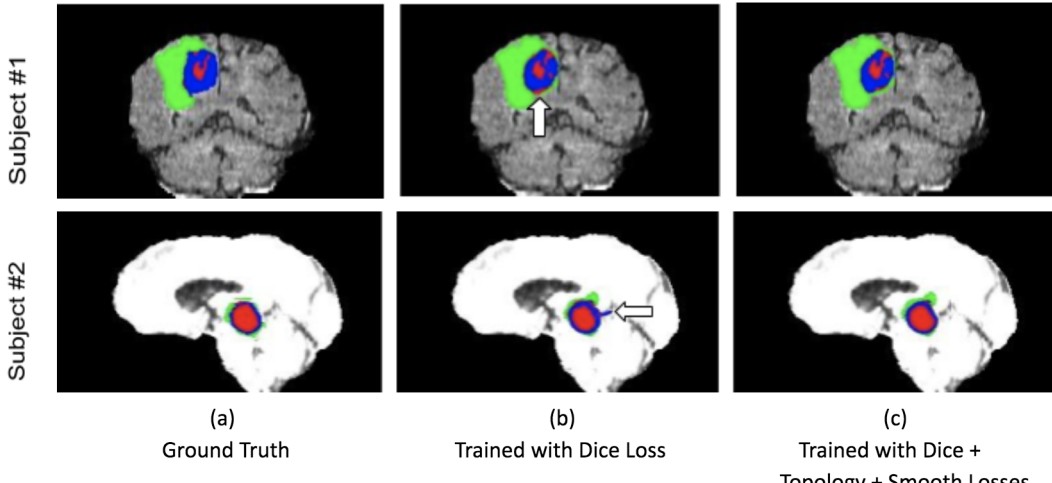

Figure 2: This figure shows two different subjects in two different views. Topology is being violated in (b) by the ET area (red) in Subject #1 and by the TC area (blue) in Subject #2. The topological loss (c) attenuates such violations.

Table 1 shows an increase in both Dice scores and Hausdorff95 distances in ET regions. The use of the topological loss improves, therefore, the segmentation accuracy in one of the most challenging areas of a brain tumor. ET regions have indeed the smallest number of voxels compared to other sub-regions (Figure 2). The improvements due to the use of a topological loss are, however, attenuated in the other tumor regions since they have a much larger number of voxels.

Table 1: Evaluation of different combinations of losses in FCNs, measured in terms of Dice scores and Hausdorff95 distances (mm).

| | Dice score | | | Hausdorff95 distance | | |
|---|---|---|---|---|---|---|
| Loss | ET | TC | WT | ET | TC | WT |
| $L_{Dice}$ | 0.511 | 0.771 | 0.875 | 8.088 | 7.660 | 13.858 |
| $L_{Dice} + L_{Topo}$ | **0.540** | 0.761 | 0.877 | **6.174** | 7.850 | 13.870 |
| $L_{Dice} + L_{Topo} + L_{Smooth}$ | **0.553** | 0.761 | 0.877 | **6.306** | 7.738 | 13.915 |

## 4. Conclusion

Our experiments reveal that the use of a topological loss (BenTaieb and Hamarneh, 2016) yields improvements in accuracy when segmenting enhancing tumors. Such topological loss could be further applied in segmentation of other structures where topological apriori is known. Code is also provided[1].

---

1. Code available at https://github.com/charan223/Brain-Tumor-Segmentation-using-Topological-Loss

## Acknowledgements

This work was supported financially by the Mathematics of Information Technology and Complex Systems (MITACS) Globalink Internship Program, and the Natural Sciences and Engineering Research Council of Canada (NSERC). We also gratefully acknowledge the support of NVIDIA Corporation with the donation of a Titan Xp GPU used for this research.

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
