# OpenReview forum: "Brain Tumor Segmentation using Topological Loss in Convolutional Networks"
_MIDL.io/2019/Conference/Abstract — MIDL Abstract 2019_

### Official Review · AnonReviewer2 · 2019-04-29
**Attempt to improve segmentation by adding specific terms to the loss function, experiments are not very convincing**

**Rating:** 3
**Confidence:** 3

**Review:**

This paper shows that a segmentation network trained with a cost function with additional terms for "topological loss" and smoothness may result in better segmentations. The experiments are not convincing. They have only been carried out on a single small dataset and only for the class "enhancing tumor" the results improve, while for the other classes the results deteriorate. No statistical analysis is done. It would be nice if the resolution of the figure could be improved. The results seem handpicked. There is no discussion about using simple post-processing rules such as taking the largest component, or using adversarial training to impose shape constraints. Although I think the work is solid, and adding terms to a loss function is an important way to improve the performance of networks, I did not rank this abstract very high among the ones on my list.

---

### Official Review · AnonReviewer1 · 2019-04-30
**Practical paper**

**Rating:** 3
**Confidence:** 3

**Review:**

This abstract investigates the use of a topological loss and shows how it can improve segmentation algorithms for tumor segmentation. The results and methods are sufficiently explained.

---

### Decision · Program_Chairs · 2019-05-06
**Acceptance Decision**

Accept